# An Anatomically Preserved Cone-like Flower from the Lower Cretaceous of China

**DOI:** 10.3390/life13010129

**Published:** 2023-01-03

**Authors:** Xin Wang, José B. Diez, Mike Pole, Manuel García-Ávila

**Affiliations:** 1State Key Laboratory of Palaeobiology and Stratigraphy, Nanjing Institute of Geology and Palaeontology and CAS Center for Excellence in Life and Paleoenvironment, Chinese Academy of Sciences, Nanjing 210008, China; 2Departamento de Xeociencias Mariñas e Ordenación do Territorio, Universidade de Vigo, 36200 Vigo, Spain; 3Centro de Investigación Mariña, Universidade de Vigo (CIM-UVIGO), 36200 Vigo, Spain; 4Queensland Herbarium, Mount Coot-Tha Road, Toowong, QLD 4066, Australia

**Keywords:** flower, cone, gymnosperms, angiosperms, evolution, Cretaceous, China

## Abstract

Although diverse fossil angiosperms (including their reproductive organs) have been reported from the Early Cretaceous, few of them are well-documented due to poor preservation and limited technologies available to apply. For example, paraffin sectioning, a routine technology applied to reveal the anatomical details of extant plants, was hitherto at most rarely applied to fossil plants. This undermines the comparability between the outcomes of studies on fossil and extant plants, and makes our understanding on plants incomplete and biased. Here, we applied paraffin sectioning technology, in addition to light microscopy, SEM, and TEM, to document a fossil reproductive organ, *Xilinia* gen. nov., from the Early Cretaceous in Inner Mongolia, China. The anatomical details of this new reproductive organ were documented. *Xilinia* bears a remarkable resemblance to conifer cones, although its ovules are enclosed in carpels. The paradoxical cone-like morphology of *Xilinia* appears to represent a transitional snapshot of plant evolution that is absent in extant plants.

## 1. Introduction

Angiosperms are by far the most important plant group for humans, as they provide most of the necessary materials for the origin and survival of humans. Naturally, angiosperms are one of the foci for botanical studies. Among all the questions concerning angiosperms, the origin of angiosperms is a core question of key significance since answers for many other questions are hinged upon the answer to this question. Previously, confronting hypotheses were raised to account for the origin and radiation of angiosperms in the Cretaceous, for example, Euanthial theory vs. Pseudoanthial theory, and the phyllosporous school vs. the stachyosporous school. Although Euanthial theory used to be of great influence in the 20th century due to the pioneering work of Arber and Parkin [1], now both the Euanthial and Pseudoanthial theories are out of phase due to the lack of fossil evidence favoring either of them, while the previously confronting phyllosporous and stachyosporous schools seem to have been reconciled by the recently advanced Unifying Theory, which is mainly based on fossil evidence [2]. In the past three decades, paleobotanists have uncovered many interesting discoveries of early angiosperms from the Early Cretaceous (including *Chaoyangia* [3], *Archaefructus* [4,5,6,7], *Sinocarpus* [8,9], *Callianthus* [10,11], *Liaoningfructus* [12], *Sinoherba* [13], and *Varifructus* [14]), which shed new light on this question [2,3,4,5,6,7,8,9,10,11,12,13,14,15,16,17,18,19]. However, from which group angiosperms were derived remains an unanswered question [20,21,22,23,24], partially due to a lack of anatomically preserved fossils of related plant reproductive organs that can be compared with extant plants. Here, we report a fossil female flower, *Xilinia shengliensis* gen. et sp. nov, which demonstrates angio-ovuly (a characteristic feature of angiosperms) and cone-like morphology (an unusual feature in angiosperms). Its chimeric character assemblage makes *Xilinia* unique and informative for flower origin and evolution.

## 2. Materials and Methods

The Lower Cretaceous of Inner Mongolia, China is famous for its abundant coal resources. The coal-bearing strata in the Xilinhot region belong to the Baiyanhua Group, which is divided into the overlying Shengli Formation and the underlying Xilin Formation. The Shengli Formation in the Baiyanhua Group was also called the Saihantala or Saihan Tal Formation in publications by various authors [25]. Stratigraphically, the Shengli Formation is comparable to the Fuxin and Chengzihe Formations in Northeast China. The Shengli coal mine near Xilinhot, Inner Mongolia, China has been exploring the coal resources in the Shengli Formation for industrial usage. Coal layers in the formation were named by increasing numbers in a top-to-down order. Our fossil material was collected from siltstones overlying Coal Layer 5 in the Shengli coal mine in 1994 (44°0′20″ N, 116°1′10″ E, Figure 1a–d). Recently, two tuffs sandwiched in Coal Layers 5 and 6 have been dated as about 107 Ma (Albian, late Early Cretaceous) using isotopic U–Pb single zircon grain dating via LA–ICP–MS [26].

The Shengli Formation mainly comprises mudstones, shales, fine-grained sandstones, and conglomerates intercalated with oil shales and coal beds [27]. Various fossils, including brackish-water dinocyst, conchostracans, ostracods, gastropods, bivalves, and fish, were reported in the formation [25,27,28,29]. Its dinocyst assemblage is characterized by index genera *Nyktericysta* and *Vesperopsis*. Its conchostracans (*Eosestheria*), gastropods (*Probaicalia gerassimovi*), bivalves (*Arguniella* sp., *Sphaerium* (*S.*) *anderssoni* and *Sphaerioides yixianensis*) are comparable to those in the Jiufotang and Fuxin Formations [30].

Our specimen is coalified compression, embedded in a block about 12.5 × 11 × 4 cm. Morphological details were observed using a Hitachi S800 SEM at the Institute of Botany, CAS. The axis and some of the carpels of the fossil were dislocated and treated with pure HNO_3_ before paraffin sectioning. The treatment was stopped when the specimen became more or less elastic. Then, the sample was embedded in paraffin and sectioned at an interval of 8 μm after Johansen’s [31] and Jensen’s [32] guidance, which was designed for extant plant materials. It was paraffin-sectioned and photographed using a Nikon 4500 microscope. A fragment of an ovule was embedded in SPI-pon 812, and sectioned using a Leica Ultracut R ultramicrotome set at a 70 nm interval using a diamond knife at Nanjing Normal University, Nanjing, China. The ultrathin sections were stained with lead citrate at 1/60 of the standard concentration, and observed using a Hitachi-7650 TEM at Nanjing Normal University. All photographs were organized for publication using Photoshop 7.0.

## 3. Results

**Class:** Angiosperms.

**Order:** Incertae Sedis.

**Family:** Incertae Sedis.

**Genus:***Xilinia* gen. nov.

**Generic diagnosis**: Female organ cone-like, pedicellate, with helically arranged carpels and perianth. Carpel composed of an ovary wall and an anatropous ovule. Ovular membrane sac-like with an extra layer in the distal portion, three-layered.**Type species**: *Xilinia shengliensis* gen. et sp. nov.**Etymology**: *Xilin*, for the city of Xilinhot where the fossil was collected.

*Xilinia shengliensis* gen. et sp. nov.

**Species diagnosis**: (In addition to that of the genus), organ up to 23 mm long and 14 mm wide. Pedicel up to 3.5 mm in diameter. Perianth element linear, narrow, up to 22 mm long. Gynoecium oval, about 1 cm in diameter, with multiple carpels. Carpel inverted triangular, up to 2.3 mm long and 1.1 mm wide, three-dimensionally preserved, papillate apically. Ovule single in a carpel, anatropous, with sac-like ovular membrane of amber color.

**Description**: Only one specimen (holotype) was collected for our study. The parts of distinct morphologies aggregated in our fossil eliminate the possibility of its being an inflorescence, which is expected to have flowers of relatively uniform morphology. The flower is female, pedicellate, with perianth, up to 23 mm long and 14 mm wide (Figure 2a,b and Figure 3a). The pedicel is about 3.5 mm in diameter, 11 mm long (Figure 2a–f). The perianth is composed of numerous linear perianth elements that are rhomboidal in cross-view, leaving rhomboidal scars on the axis, 1 mm wide, at least 22 mm long, with simple pits on the walls of tracheids (Figure 2a–g and Figure 4a,b). The gynoecium is ovoid, about 8 mm wide and 9 mm long, with numerous carpels helically arranged on the flower axis (Figure 2a,b, Figure 3a and Figure 5a,f). The axis is about 3.5 mm wide in the bottom, tapering distally, with numerous stubs left by fallen carpels on it (Figure 2a,b, Figure 4e,h and Figure 6a,b). Anatomically, the axis is composed of a central pith surrounded by secondary xylem, phloem, and epidermis (Figure 6a–i). The pith is parenchymatic, about 1.1 mm in diameter (Figure 6a,f–h). Cells in the pith are more or less isodiametric in cross-view, 17–25 × 19–39 μm, elongated longitudinally, 49–100 μm long, some with simple pits on their cell walls and organic infillings in the lumina (Figure 6a,c,f–h). There are two types of organic infillings, solid or spongy, in the cell lumina (Figure 6c,d). The xylem includes isolated primary xylem bundles and a secondary xylem cylinder (Figure 6f–h). The primary xylem is endarch and distributed along the margin of the pith (Figure 6f–h). The secondary xylem forms a ring around the pith, is about 0.7 mm thick, composed of tracheids, cavities, and rays, penetrated by carpel traces (Figure 6a,b,f). The tracheids are about 18 μm wide with simple and bordered pits (Figure 5i,j and Figure 6c,d,g,h). The cavities frequently seen in the early secondary xylem are up to 265 μm long and 115 μm in diameter, probably lysigenous (Figure 6a,b,d–f,i). Rays are uniseriate, 3–8 cells high, and up to 72 μm high (Figure 6e). The phloem and epidermis of the flower axis were hardly discernable in the paraffin sections, probably due to the nitric acid processing, although the epidermis could be seen using SEM (Figure 4e,h). There was no trace of any androecium in the flower (Figure 2a,b). More than 60 carpels are helically arranged along the flower axis, and the traces of these carpels penetrate the xylem cylinder in the flower axis (Figure 2a,b, Figure 3a and Figure 6b,f,g). Carpel size and shape vary depending on their positions in the gynoecium: they are inverted triangular in adaxial and abaxial views, wedge-shaped in side view, 1.7–2.3 mm long, 0.7–1.06 mm wide, and 0.56–0.85 mm thick (Figure 3c, Figure 4c,d,f,g and Figure 5a,f). A carpel comprises an ovary wall and an anatropous ovule within (Figure 1c–h, Figure 3c, Figure 4c and Figure 5f–h). The ovary wall is composed of longitudinally oriented hypodermis and epidermis (Figure 5a,b,d–f,h and Figure 6j). The epidermal cells are isodiametric in surface view, about 7–12 × 11–22 μm in surface view (Figure 6j). The ovary wall may be up to 35 μm thick, with longitudinal striations on its inner surface (Figure 5d–h). There is no style, and the papillae are restricted to the distal portion of the carpel (Figure 5a–c). The ovule is anatropous, with its micropyle close to the flower axis (Figure 5f–h). The integument encloses nucellus, up to 83 μm thick, composed of radially arranged parenchymatic cells, easily dissolved in nitric acid (Figure 5f–h). The ovular membrane sac is about 1.8 mm long, 1 mm wide, thin, smooth-surfaced, amber in color, tapering distally, of longitudinally oriented cells, becoming thicker at the apex due to additional layer of the integument, enclosed by the integument (Figure 3c–e, Figure 4c, Figure 5f,h, Figure 6k–m and Figure 7a–d). Little content is seen within the ovular membrane; therefore, two layers of the membrane are tightly compressed against each other (Figure 7a–c). Each membrane is about 1.7 μm thick, including three distinct layers, namely, a 0.36 μm thick foot layer, a 0.86 μm thick columella layer, and a 0.66 μm thick tectum layer (Figure 7a–g). The columella layer includes sparse rod-formed vertical structures separated by wide space (Figure 7e–g). The tectum layer covers the columella layer with some stratification (Figure 7e–g).

**Etymology**: *shengli*, for the name of the formation from which the fossil was collected.**Holotype specimen**: 9222.**Depository**: Institute of Botany, Chinese Academy of Sciences, Beijing, China.**Age and horizon**: the Albian, Early Cretaceous; the Shengli Formation.**Locality**: Xilinhot, Inner Mongolia, China.

## 4. Discussion

A strict and sufficient criterion for angiosperms is angio-ovuly, that is, the ovules being enclosed before pollination [2,33]. All plants with their ovules enclosed before pollination are unexceptionally angiosperms. The presence of an ovular membrane with little content in *Xilinia* suggests that the original content lacked fossilizable materials before fossilization. Considering that the very delicate parenchyma of the integument has been preserved perfectly in the same fossil (Figure 5f–h), such a lack of preserved material within the ovular membrane sac implies that the ovules of *Xilinia* were still premature, lacking cellularized content when fossilized. Therefore, *Xilinia* was in its pre-pollination stage when fossilized. The ovule is inside the ovary wall that is integral except for physical cracks caused by desiccation (Figure 2c,d,f,g and Figure 5a,f,g), suggesting that the ovules of *Xilinia* are fully enclosed by their ovary walls. This enclosure of ovules before pollination is in line with the occurrence of papillae on the carpel tip (Figure 5a–c), which may function as a stigma during the pollination. All the above information collectively points to the fact that *Xilinia* is an angiosperm, and allows for us to reconstruct it as shown in Figure 8.

Although it is an angiosperm, *Xilinia* has several features frequently seen in gymnosperms and unexpected for typical angiosperms, including ovular membrane with columellate stratification, lack of a typical perianth, unisexuality, and bordered pits. We discuss these characters and their implications.

The occurrence of ovular membranes is rare in angiosperms [34] and only poorly developed in Magnoliaceae [17], so the presence of an ovular membrane in *Xilinia* appears at odds with its angiosperm affinity. Furthermore, the ovular membrane stratification present in *Xilinia* has never been seen in any known seed plants [17,35,36,37,38,39,40,41,42,43,44,45,46,47,48,49]. Ovular/megaspore membranes have been reported in various fossil taxa, but only a limited number of them were examined using TEM, making our comparison here incomplete. Among those observed using TEM, none of Cladoxylopsids [48], Lycopodales [38,39,48,50,51,52], Isoetales [53], Marsileales [42], or those of Pteridospermales [35,52], Cordaitales [52], Ginkgoales [37,40,46], and Coniferales [43,54] has a wall stratification comparable to that seen in *Xilinia*. Among them, only some Pteridospermales, Cordaitales, and Lycopodales [35,52], Cordaitales [52], Ginkgoales [37,40,46], and Coniferales [43,54] have a three-layered ovular/megaspore membrane stratification. However, their spongy layer is composed of anastomosing bacula and is thus distinct from the vertically oriented, distantly spaced rod-like structures seen in the columella-like layer of *Xilinia* (Figure 7e–g). Thus, *Xilinia* is unique among seed plants in terms of ovular membrane organization. Such a stratification is comparable to that of a typical angiosperm pollen wall.

The helical arrangement of the carpels in *Xilinia* is not only like that of carpels in Magnoliales [55], but also like that of lateral cone appendages in Cycadales, Bennettitales, and Coniferales [56,57,58]. Such a feature has been taken as a plesiomorph in angiosperms [59].

The perianth elements of *Xilinia* are linear or needle-like rather than petaloid, implying that these elements may not yet play protective or attractive roles as their counterparts do in typical extant flowers. This interpretation agrees with the lack of a typical perianth in early angiosperms (*Chaoyangia* [3], *Archaefructus* [4,5,6,7], *Sinocarpus* [8,9], *Callianthus* [10,11], *Eofructus* [60], etc.) and gymnosperms, although perianth may have a history tracing back to the Middle–Late Jurassic according to the latest progresses [61,62].

Dioecism is a feature frequently seen in gymnosperms, but only relatively rarely seen in angiosperms [56,58,59,63,64]. In this respect, *Xilinia* appears to be more similar to cones in conifers rather than a typically bisexual flower. This comparison is further underscored by the needle-like morphology of the perianth elements in *Xilinia*.

The vessel element in xylem is frequently used as an anatomical feature to identify angiosperms, although it occurs in some living Gnetales [65] and fossil Giantopteridales [66,67]. There are cavities in the early secondary xylem in the flower axis of *Xilinia*. These cavities are up to 265 μm long and 115 μm in diameter, far beyond the dimensions of a regular tracheid and more comparable to a vessel element (Figure 6a,b,e,i). The possibility of them being resin ducts is slim, as resin content has been seen in the same axis, but is lacking in these cavities. Since we did not identify a perforation plate characteristic of vessel elements, and they appeared to not be connected to each other, we are not sure whether they represent vessel elements. If they do represent vessel elements, they might stand for the initial development of vessel elements in which these isolated cavities were yet to develop through lysigeny into a full vessel. Otherwise, the function of such cavities is mysterious.

The presences of simple pits (frequently seen in angiosperms) in the vascular bundle of the perianth element and bordered pits (frequently seen in gymnosperms, but only rarely seen in Monocots, e.g., *Dracaena* [68]) in the flower axis of *Xilinia* suggests an affinity swaying between angiosperms and gymnosperms, in line with the cone-like morphology and organization of *Xilinia*. This explains why we hesitated on placing *Xilinia* in a specific order or family in angiosperms. In short, *Xilinia* seems to have not completed its transition from gymnosperms to angiosperms; thus, it bridges these otherwise distinctly separated groups.

## 5. Conclusions

*Xilinia* is a new reproductive organ of a fossil plant from the late Albian, Early Cretaceous. The fossil’s anatomical details were well-revealed and -documented due to the application of paraffin sectioning, which was rarely applied on fossil plants before. *Xilinia* is unique in that it bears a remarkable resemblance to conifer cones, although its ovules are enclosed in carpels. Its cone-like flower morphology appears to represent a transitional snapshot of plant evolution that is now missing in extant plants.

## Figures and Tables

**Figure 1 life-13-00129-f001:**
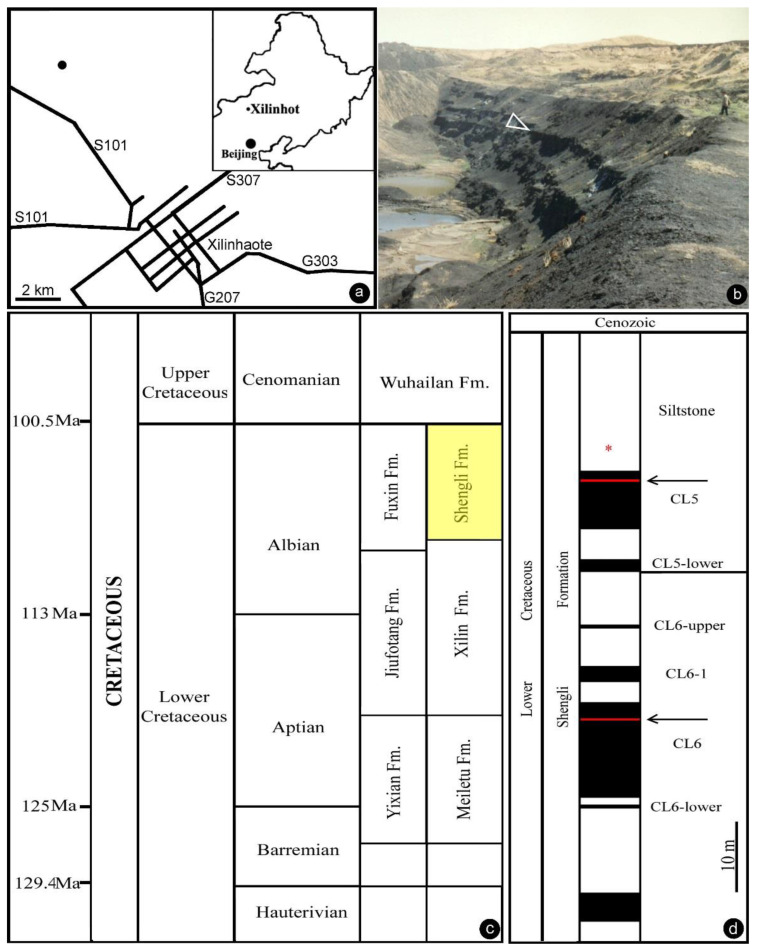
General information of the fossil locality. (**a**) Geographical position of the fossil locality. The inset shows Northeast China. The main map shows the fossil locality (black dot) in the northwest suburb of the city of Xilinhot. (**b**) Outcrop at the Shengli coal mine near Xilinhot, Inner Mongolia, China (44°0′20″ N, 116°1′10″ E) in 1994. The fossil specimen was collected from the arrowed layer. (**c**) Stratigraphy in Northeast China. Note the horizon of the Shengli Formation (yellow) in the Albian, upper Lower Cretaceous. Modified from Shen et al. [26]. (**d**) Stratum column of the Shengli coal mine showing the horizon yielding our fossil (asterisk). Modified from Shen et al. [26].

**Figure 2 life-13-00129-f002:**
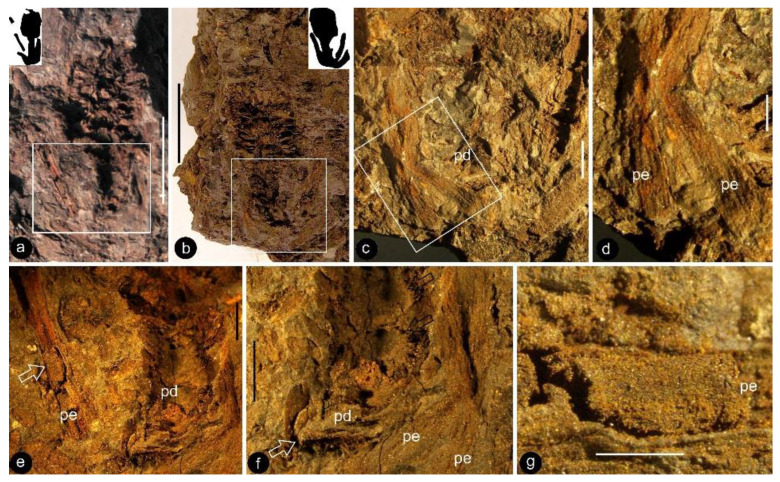
*Xilinia* gen. nov. and its details. Light microscopy. Specimen number 9222. (**a**,**b**) Two facing parts showing the cone-like morphology of the organ. The organ was longitudinally split through the middle; shown here are longitudinal views. Note the helical arrangement of the carpels around the axis (missing in the figures). Scale bar = 1 cm. (**c**) Detailed view of the rectangle in (**b**), showing the pedicel (pd) and surrounding perianth elements. Scale bar = 2 mm. (**d**) Detailed view of two perianth elements (pe) in (**c**). Scale bar = 1 mm. (**e**) Detailed view of the rectangle in (**a**), showing the pedicel (pd) and one of the surrounding perianth elements (pe, arrow). Scale bar = 2 mm. (**f**) Detailed view of the bottom-right portion in (**e**) showing the pedicel (arrow, pd) and physically connected surrounding perianth elements (pe). Scale bar = 2 mm. (**g**) Detailed view of the perianth element (pe) arrowed in (**e**) showing the organic preserved tissue. Refer to Figure 4a,b. Scale bar = 1 mm.

**Figure 3 life-13-00129-f003:**
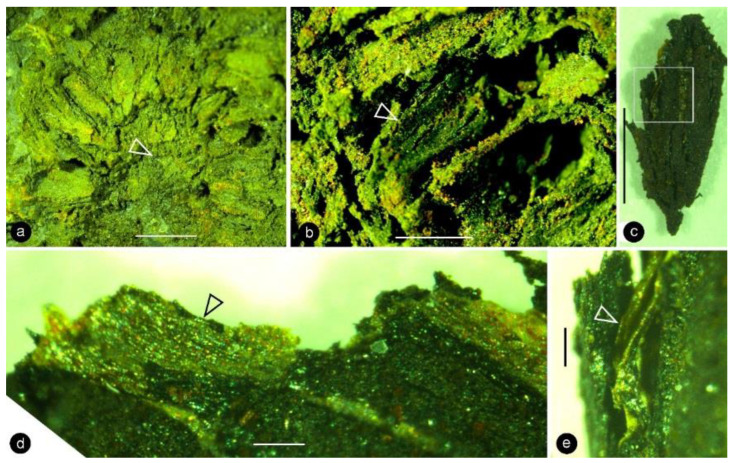
Details of the carpels and ovular membrane. Light microscopy. Specimen number 9222. (**a**) Multiple carpels helically arranged around the center (arrow). Scale bar = 2 mm. (**b**) One of the carpels (arrow) still embedded in the sediments. Scale bar = 1 mm. (**c**) The same carpel as that in (**b**), dislocated from sediments. Scale bar = 1 mm. (**d**) Amber-colored ovular membrane (arrow) *in situ* in a carpel. Scale bar = 0.1 mm. (**e**) Shiny amber-colored ovular membrane (arrow) embedded in coalified tissues, enlarged from the rectangle in (**c**). Scale bar = 0.1 mm.

**Figure 4 life-13-00129-f004:**
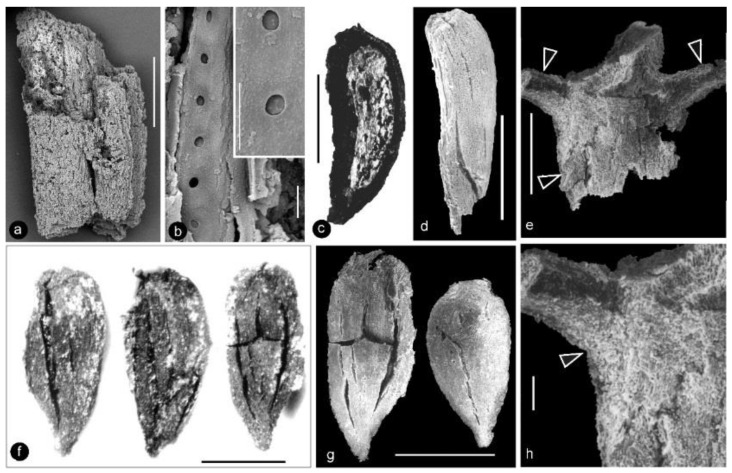
Details of perianth elements, carpels, and flower axis of *Xilinia* gen. nov. SEM except (**c**,**f**), which are stereomicroscopy. Specimen number 9222. (**a**) A fragment of a perianth element shown in Figure 2g. Scale bar = 5 mm. (**b**) Simple pits on the tracheid wall in the perianth element shown in (**a**). Two of them are shown in detail in the inset. Bars are 5 and 1 μm, respectively, long. (**c**) A side view of a broken carpel, of the same orientation as that in (**d**). Note the light-colored ovular membrane inside the carpel. Scale bar = 1 mm. (**d**) A side view of a wedge-shaped carpel. Scale bar = 1 mm. (**e**) A portion of the axis with helically arranged stubs (arrows) left by the fallen carpels. Scale bar = 0.5 mm. (**f**) Adaxial view of three carpels of triangular shape. Scale bar = 1 mm. (**g**) Abaxial view of two carpels of variable triangular shapes. The right one is from the distal portion of the gynoecium. Note the cracks due to desiccation. Scale bar = 1 mm. (**h**) Details of the axis shown in (**e**). Note the integral surface (arrow) below the carpel stub, suggestive of a lack of subtending bract expected in a conifer cone. Scale bar = 0.1 mm.

**Figure 5 life-13-00129-f005:**
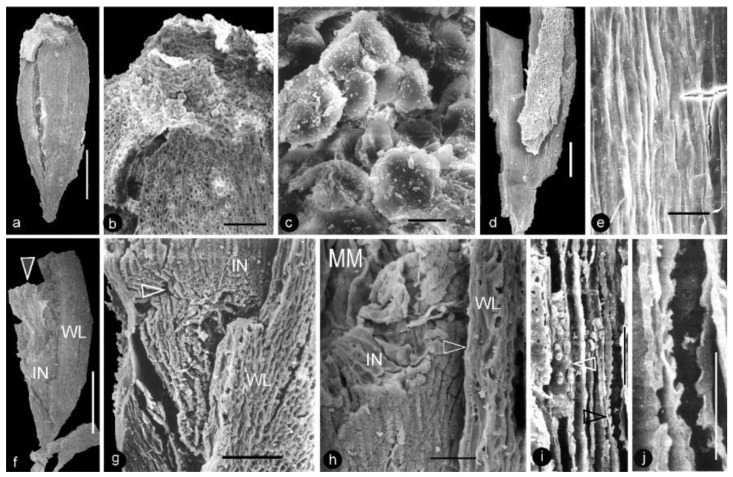
Morphology and anatomy of carpels. SEM. Specimen number 9222. (**a**) Adaxial view of an integral carpel of triangular shape. Note the cracks due to desiccation. Scale bar = 0.5 mm. (**b**) Apex of the carpel in (**a**). Note the cellular details on the carpel surface (lower) and papillae on the apex (upper). Scale bar = 0.1 mm. (**c**) Detailed view of the papillae on the apex of the carpel in (**b**). Scale bar = 10 μm. (**d**) Ovary wall with cellular details. Scale bar = 0.2 mm. (**e**) Longitudinal striations on the inner surface of the ovary wall, enlarged from (**d**). Scale bar = 30 μm. (**f**) A broken carpel showing ovary wall (WL), integument (IN), and *in situ* ovular membrane (arrow). Scale bar = 0.5 mm. (**g**) Detailed view of the lower portion of the carpel in (**f**) showing the cellular details of the ovary wall (WL), integument (IN), and micropyle (arrow). Scale bar = 0.1 mm. (**h**) Detailed view of the middle portion of the carpel in (**f**), showing the cellular details of the ovary wall (arrow, WL), radial files of parenchymatic cells in the integument (IN), and ovular membrane (MM). Scale bar = 50 μm. (**i**) Anatomy of the flower axis showing the pith parenchyma with organic infilling (white arrow) and bordered pits on the tracheid wall (black arrow). Scale bar = 30 μm. (**j**) Detailed view of bordered pits on the tracheid wall enlarged from (**i**). Scale bar = 20 μm.

**Figure 6 life-13-00129-f006:**
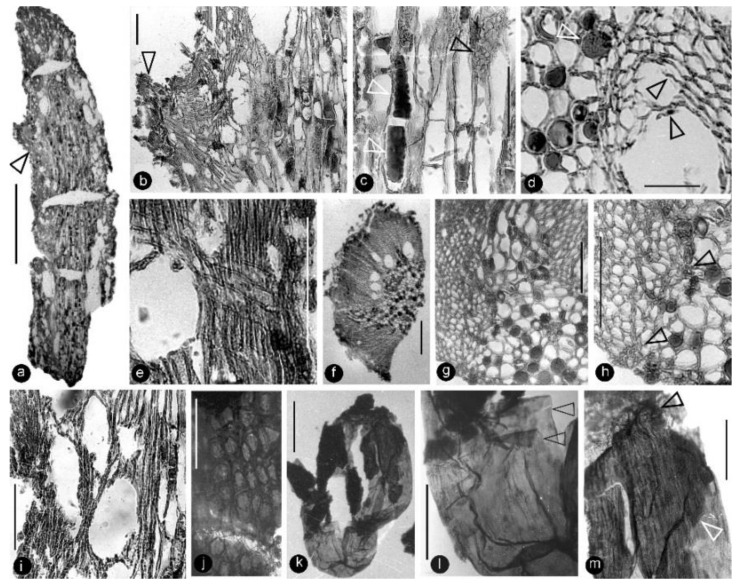
Anatomy of the flower axis and details of the ovular membrane. Specimen number 9222. Light microscopy. (**a**) A longitudinal section of the flower axis. Scale bar = 1 mm. (**b**) Detailed view of the axis, enlarged from the arrowed region in (**a**), showing a trace (arrow) to a carpel and pith (right). Scale bar = 0.1 mm. (**c**) Detailed view of the pith enlarged from (**a**) showing the elongated rectangular cells with two types of organic infillings (lower white arrow and black arrow) and simple pit (upper white arrow) on the cell wall. Scale bar = 0.1 mm. (**d**) A cross-view of the pith (left) and adjacent xylem showing two types of infillings in the parenchymatic cells in the pith (white arrow) and residual cell walls (black arrows) in the lysigenous cavity in the xylem. Scale bar = 50 μm. (**e**) A longitudinal view of the xylem in flower axis, showing the cavity (left) in the xylem and a cross field formed by a three-cell-high ray and tracheids in the xylem. Scale bar = 0.1 mm. (**f**) A cross-section of a portion of the flower axis showing cavities in the inner portion of the xylem. Scale bar = 0.2 mm. (**g**) A detailed cross-view of the flower axis showing the pith (lower right), xylem (upper left), a trace to a carpel (middle top), and some cells in pith with organic infillings. Scale bar = 0.1 mm. (**h**) Another detailed cross-view of the flower axis showing the pith (right), xylem (left), small cells of the protoxylem (arrows), and some cells in the pith with organic infillings. Scale bar = 0.1 mm. (**i**) Semilongitudinal section of the axis, showing pith (right) and xylem (left) with cavities. Scale bar = 0.1 mm. (**j**) Elongated or isodiametric epidermal cells of an ovary wall. Scale bar = 50 μm. (**k**) A whole ovular membrane with some coalified integument attached. Scale bar = 0.5 mm. (**l**) A detailed view of the proximal portion of the ovular membrane shown in (**k**). Note the transparent amber-like color and two layers of the ovular membrane (arrows). Scale bar = 0.2 mm. (**m**) Apical portion of the ovular membrane shown in (**k**), showing the elongate cells on the membrane and an additional layer (arrows), probably due to the presence of integument. Scale bar = 0.2 mm.

**Figure 7 life-13-00129-f007:**
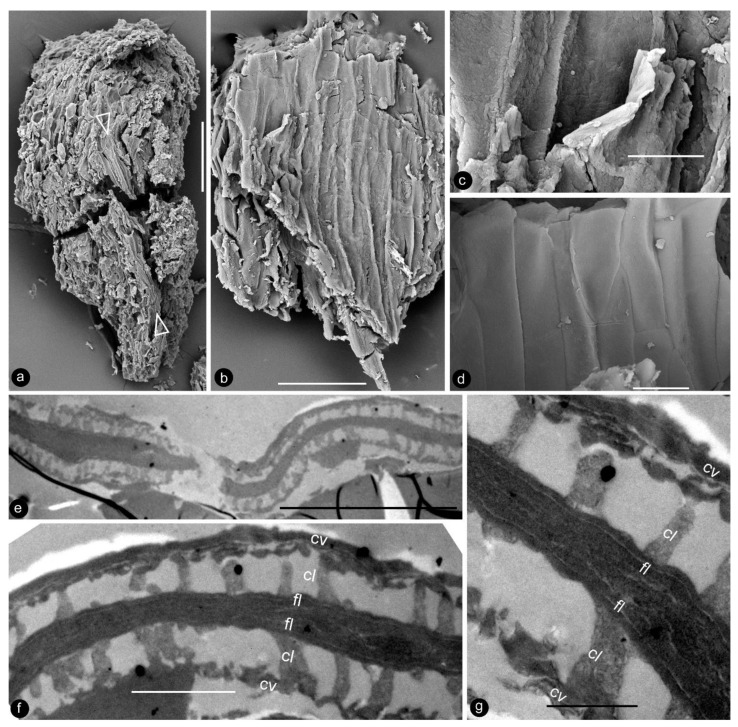
Details of the ovular membrane. Specimen number 9222. (**a**–**d**) are SEM, (**e**–**g**) are TEM. (**a**) A longitudinal view of an ovule fragment. Note the ovular membrane (arrows) sandwiched between parenchymatic integument tissues. Scale bar = 0.1 mm. (**b**) External surface view of an ovular membrane, showing elongated cell outlines. Scale bar = 50 μm. (**c**) Detailed view of (**b**), showing two layers of the membrane appressed each other. Scale bar = 10 μm. (**d**) Internal surface of ovular membrane showing elongated cell outline. Scale bar = 10 μm. (**e**) Two layers of the ovular membrane shown in (**a**), appressed against each other. Scale bar = 10 μm. (**f**) Detailed view of (**e**), showing two appressed foot layers (fl), columella layer (cl), and tectum layer (cv). Scale bar = 2 μm. (**g**) Detailed view of the ovular membrane enlarged from (**f**), showing two compressed foot layers (fl), columella layer (cl), and tectum layer (cv). Note the light-colored lines in the foot layer. Scale bar = 1 μm.

**Figure 8 life-13-00129-f008:**
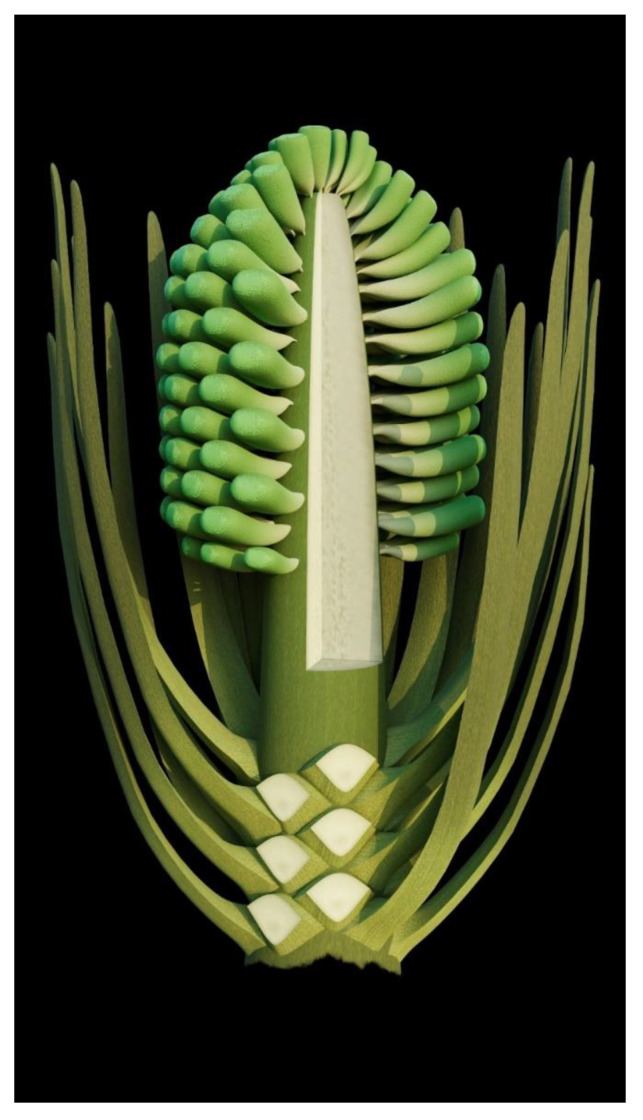
Reconstruction of *Xilinia*.

## Data Availability

All data are reported in this paper.

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
