# Peer review of "An Anatomically Preserved Cone-like Flower from the Lower Cretaceous of China"

_life, 2023, doi:10.3390/life13010129_

Round 1
Reviewer 1 Report
This is an interesting paper presenting a new cone-like flower from the Albian of the Shengli Coal Mine (China). I have some suggestions that I hope will be helpful. Please see details bellow:
-L43. “the provenance of angiosperms remains a focus of debate”: do you mean from the geographical/stratigraphical/environmental point of view ?
L51. The geological context is maybe too light. Specify the stratigraphic context of the “Coal Layer 5”. Any stratigraphic section of the quarry ? Lithology and depositional environment of the plant bearing bed as well as the rest of the series exposed in the quarry ? Any reference concerning sedimentology of the quarry?
L52. You indicate an Albian age, please specify what are the dating arguments.
L72. The type material need to be specify. Any paratype ? You give a generic diagnosis however the diagnosis of the new species is missing. Needs to be update.
L75. Figure 2: The resolution of the photographs is quite limited. In fig1a and fig1b, sketches could be helpful for the reader. In the caption, indicate the type of views.
L91. Caption of figure 3: the specimen number needs to be indicate.
L99. “embedded in a block about 12.5 cm […]” could be moved in the material section (do not concern the description of the flower.
L100. Any argument to exclude that the specimen consists of an inflorescence ?
L102. Any stomata on the perianth surface ?
L103. “linear, rhomboidal in cross view, leaving rhomboidal scars on the axis” strongly looks like the morphology of needle leaves present amongst common Cretaceous Conifers (e.g. Cunninghamites, Geinitzia, Elatocladus).
L104. The figures do not appear in the order in the text. Please update.
L120. Caption of figure 4: specimen number needs to be indicate. Same in other captions.
L.163. “their traces”, what do you mean ?
L168. Dimensions of hypodermal cells?
L.189. You missed to explain the etymology of the genus and species names.
L.300. “Xinlinia” > “Xilinia”, check everywhere.
Reviewer 2 Report
It is a well-done paleobotanical study, and I like it very much. I would add some more data on similar recent taxa with an additional illustration, but it is not necessary.
Reviewer 3 Report
Lines 28-29. Documented is used twice in one sentence. How about :…have been reported in the Early Cretaceous…..
Introduction – not all readers are familiar with fossil plants and angiosperms. Angiosperms is important plant groups and maybe some additional information about their origin, history of research, the most important contributions, main morphological/anatomical characteristics etc would be needed.
It need not be long
Line 52 …Coal Mine, Inner Mongolia, China.
Some brief summary of the geological situation – Geological settings – thatś also routine way for every taxonomic manuscript. Maybe including graphical form/table/scheme of the stratigraphy?
3. Results
It´d be good to locate a new genus into the systém. Yes, i tis transitional form, so maybe Incertae Sedis?
Erection of a new taxon has some routine information including Type species, Type locality, Stratigraphy, Derivation of name, Diagnosis for a new genus and the same including Holotype and Description for a new species.
344 end of reference : et al??
346 p. 347 or pp. 347?
350 dtto p or pp.?
411 dtto
Reference N. 20 probably is not mentioned in the text.
